# NF1 with Multiple Cardiac Structural Abnormalities Leading to Cerebral Infarction

**DOI:** 10.3390/diagnostics16010163

**Published:** 2026-01-04

**Authors:** Jingwei Ye, Yiyi Jiang, Hanmin Wang, Dan Wang

**Affiliations:** 1Department of Neurology, The First Affiliated Hospital of Wenzhou Medical University, Wenzhou 325035, China; yjw@wmu.edu.cn (J.Y.); whmbb1@126.com (H.W.); 2First Clinical Medical College, Wenzhou Medical University, Wenzhou 325035, China; 3Department of Radiology, The First Affiliated Hospital of Wenzhou Medical University, Wenzhou 325035, China; jyy1106@163.com; 4Department of Pediatrics, The First Affiliated Hospital of Wenzhou Medical University, Wenzhou 325035, China

**Keywords:** neurofibromatosis type 1 (NF1), cardiac complications, valvular vegetation, thrombotic risk, cerebral infarction

## Abstract

**Background/Objectives**: Neurofibromatosis type 1 (NF1) is an autosomal dominant disorder driven by mutations in the NF1 gene, whose pathogenesis centers on the loss of neurofibromin function and subsequent hyperactivation of the RAS/MAPK pathway. Notably, to the best of our knowledge and following a systematic literature search conducted by our research team, no cases of NF1 complicated by severe cardiac structural abnormalities that ultimately lead to cerebral infarction have been reported to date. Thus, it is of paramount importance to avoid missed diagnosis by performing comprehensive cardiac-related examinations in patients with NF1. **Case Presentation**: A 20-year-old male patient diagnosed with NF1 presented with right-sided limb weakness and was initially identified with cerebral infarction. To clarify the underlying etiology, a comprehensive clinical evaluation was performed, including cardiac imaging assessments (to characterize cardiac structural changes) and whole-exome sequencing (to identify the presence of procoagulant gene mutations). Comprehensive evaluation revealed a spectrum of cardiac structural abnormalities in the patient: aortic valve prolapse with severe regurgitation, non-infective vegetations on the aortic valve leaflets, mild-to-moderate mitral regurgitation, left ventricular hypertrophy and dilation, and left atrial dilation. Whole-exome sequencing detected exclusively a pathogenic variant in the NF1 gene, with no other pathogenic/likely pathogenic variants or thrombophilia-associated polymorphisms being found. Laboratory investigations ruled out infectious etiologies, supporting the notion that NF1-mediated cardiac structural and developmental anomalies are the primary driver of cardiac vegetation formation, given the absence of other identified contributing factors; embolization of one such vegetation ultimately led to both splenic and cerebral infarction. **Conclusions**: This case emphasizes the necessity of implementing early and proactive cardiac evaluations in patients with NF1. Additionally, for NF1 individuals—particularly those presenting with suggestive vascular or cardiac symptoms—a comprehensive multifactorial assessment of thrombotic risk is critical. Collectively, maintaining clinical vigilance for cardiac abnormalities in NF1 patients and avoiding diagnostic oversight is essential to reduce life-threatening risks.

## 1. Introduction

Neurofibromatosis encompasses a group of genetic disorders classified as type 1 (NF1), type 2 (NF2), and type 3 (NF3). Café-au-lait macules and neurofibromas are hallmark features of NF1, whereas NF2 is primarily characterized by schwannomas, meningiomas, and ependymomas [1]; NF3 is characterized by multiple schwannomas typically sparing the vestibular nerves. Notably, NF1 frequently involves multiple organ systems and is associated with cardiac malformations [2,3,4].

Neurofibromatosis type 1 (NF1) is an autosomal dominant genetic disorder. This condition impairs neurofibromin function, leading to disrupted cellular growth regulation [5]. For this patient, we performed whole-exome high-throughput sequencing, which revealed a NF1 c.2889_2893del (p.Q963Hfs*10) (NM_000267.3) variant in the patient. The patient exhibited café-au-lait spots, nodular neurofibromas, and plexiform neurofibromas, which are typical clinical phenotypes of NF1.

Notably, beyond its classic tumor manifestations, NF1 is also associated with unique forms of vasculopathy. This vascular involvement is thought to occur independently of tumor compression and instead arises from structural and functional abnormalities of the vessel wall due to neurofibromin deficiency [6]. This intrinsic vascular abnormality may increase susceptibility to vascular injury and contribute to both cardiovascular and cerebrovascular complications, such as myocardial infarction or cardiac dysfunction secondary to arterial stenosis or aneurysm formation. Although NF1-associated cardiovascular complications have been reported (including congenital heart disease in 2.3% to 12.6% of NF1 subjects [3,4], vasculopathy, and hypertension), with pulmonary valve stenosis, mitral valve abnormalities, and aneurysms or stenoses of the aorta, renal, and mesenteric arteries being most common [3], the concurrence of multiple significant cardiac structural abnormalities (such as aortic valve prolapse with severe regurgitation, non-infectious valvular vegetations formation, and ventricular hypertrophy/dilation) is exceptionally rare. In particular, non-infectious vegetations as a complication of NF1 have not been reported. To our knowledge, no prior case has described a patient with NF1 presenting with this specific constellation of cardiac structural abnormalities.

## 2. Case Presentation

A 20-year-old male presented to the emergency department with acute right-sided limb weakness lasting 4 h, accompanied by an unsteady gait. He had a family history of hypertension. Further history revealed mild cognitive impairment since childhood and a previous tonsillectomy at age of six. Physical examination demonstrated right upper limb muscle strength of grade 4/5, right lower limb strength of grade 3/5. Cutaneous manifestations included multiple cutaneous and subcutaneous nodules, café-au-lait spots over the anterior chest and back (progressively increased with age), with >6 lesions present, the largest measuring > 15 mm in diameter on the right shoulder skin. Freckling was observed in the axillary and inguinal regions, and Lisch nodules were identified on the iris. Most importantly, we performed whole-exome sequencing, which revealed an NF1 gene mutation. No pathogenic or likely pathogenic variants were detected in other genes, including those associated with procoagulant function. The specific methodological details are as follows: Genomic DNA was extracted from the patient’s peripheral blood using a QIAamp DNA Extraction Kit (Qiagen, Hilden, Germany), and whole-exome sequencing (WES) was performed on the NovaSeq 6000 platform (Illumina, San Diego, CA, USA) with 150 bp paired-end sequencing using the Agilent SureSelect Human All Exon V6 capture kit (Agilent Technologies, Santa Clara, CA, USA) (coverage > 98%, depth ≥ 20×); raw data were aligned to the hg19/GRCh37 genome via BWA, with PCR duplicate removal by Picard v1.57, variant calling using GATK and the Verita Trekker Variants Detection System, annotation by ANNOVAR (2024-07-15 version), and manual inspection via IGV, followed by validation of the likely pathogenic NF1 variant using Sanger sequencing on an ABI 3730XL DNA Sequencer (Applied Biosystems/Thermo Fisher Scientific, Waltham, MA, USA) with custom primers (Figure 1). This novel NF1 variant (GRCh38/hg38: chr17:31229871_31229875; NM_000267.3: c.2889_2893del; p.Q963Hfs*10) is a 5-nucleotide deletion in exon 22, disrupting the open reading frame to produce a truncated neurofibromin. It has no assigned ClinVar ID as it is unreported in public databases (gnomAD, HGMD, ClinVar). The variant was classified as likely pathogenic [7]. Given that the patient’s parents are phenotypically normal and declined WES, and based on the available data, this variant is inferred to be de novo. To further investigate the underlying etiology, Sanger sequencing was also conducted for four core thrombophilia-related genes: *SERPINC1*, *PROC*, *PROS1*, and *SERPINE1*. Consistent with the findings of WES, no clinically significant variants were identified in these four genes. Additionally, Sanger sequencing of the *SERPINE1* gene promoter region confirmed that the patient harbored the PAI-1 4G/5G heterozygous genotype. Emergency cranial CT excluded intracranial hemorrhage, and intravenous alteplase thrombolysis was administered, resulting in partial symptomatic improvement.

Imaging and ancillary investigation results were as follows: Whole-body PET-CT showed multiple metabolically active nodules/masses in the left posterior supraclavicular region, right deep axillary area, posterior to the right first rib, and mediastinal inferior retrocardiac region, suggestive of neurofibromas. Whole-body and targeted MRI revealed plexiform neurofibromas (Figure 2a), with concurrent findings of C4–C6 vertebral body deformities and cervical kyphosis (Figure 2b–d).

Brain MRI image revealed multiple acute infarcts (hyperintense on DWI) in the left fronto-parieto-occipital lobes, and periventricular regions, with the most prominent lesion in the left frontal lobe (Figure 3a–d). A protrusion in the A2 segment of the left anterior cerebral artery and a congenital variation of the right anterior cerebral artery were identified as potential anatomical substrates for stroke. Additionally, a CT scan of the abdomen showed splenomegaly with patchy hypodensities, consistent with splenic infarction (Figure 3e).

It is worth mentioning that we also conducted examinations related to the heart. Transesophageal echocardiography confirmed aortic valve prolapse with severe regurgitation and a vegetation attached to the valve leaflet, mild-to-moderate mitral regurgitation, left ventricular enlargement with ventricular wall thickening, and left atrial enlargement (Figure 4a–f).

The patient had no history of intravenous drug abuse, high-risk sexual behavior, or dental procedures within 6 months, with well-maintained oral hygiene. No fever was reported in the 2 weeks prior to admission or during the 15-day hospitalization, and hepatitis B, hepatitis C, syphilis, and HIV yielded negative results. Three sets of blood cultures collected at distinct time points within 24 h were incubated for 6 days, with no pathogenic bacteria or fungi isolated; routine inflammatory markers including C-reactive protein (CRP), erythrocyte sedimentation rate (ESR), and procalcitonin (PCT), together with routine blood count, were within normal reference ranges, and negative Treponema pallidum particle agglutination assay (TPPA) and rapid plasma reagin (RPR) test ruled out syphilitic endocarditis. Autoantibody panel testing revealed negative antinuclear antibody (ANA, titer < 1:100), anti-double-stranded DNA antibody (dsDNA), and antiphospholipid antibodies (aPLs), with serum complement levels (C3: 1.25 g/L; C4: 0.26 g/L) within normal limits; the absence of recurrent oral/genital ulcers, uveitis, or skin lesions coupled with a negative skin prick test (SPT) excluded Behçet’s disease, and anti-streptolysin O (ASO) titer < 200 IU/mL ruled out rheumatic heart disease. Additionally, routine coagulation function tests (PT, APTT and Plasma Protein C activity) were within the normal reference range. Furthermore, digital subtraction angiography (DSA) was performed to evaluate for cerebrovascular structural abnormalities, including vascular stenosis, occlusion, malformation, and NF1-related vasculopathy—pathologies that could predispose to thrombogenesis or thromboembolism. Consistent with normal DSA findings, no such vascular abnormalities were identified, supporting the hypothesis that the cerebral infarction originated from cardiogenic thrombi derived from the non-infective cardiac vegetations.

A multidisciplinary assessment concluded that the cerebral infarction resulted from a synergistic effect of cardioembolism due to vegetation detachment. Based on all current investigative findings—including negative results for hereditary thrombophilic genetic variants, normal routine coagulation parameters, and exclusion of autoimmune/infective etiologies—the formation of cardiac vegetations appears to occur independently of other prothrombotic factors, arising primarily from multiple NF1-mediated cardiac structural and developmental abnormalities. Given the high embolic risk from the vegetation and patient’s hereditary hypofibrinolytic state, full-intensity anticoagulation was indicated. However, the bleeding risk from neurofibromas required cautious dose adjustment. Bridging therapy with warfarin and low-molecular-weight heparin was initiated, with a target INR maintained at 1.8–2.5, as per clinical pharmacy consultation. Sacubitril/valsartan was administered for cardiac remodeling and blood pressure control, and atorvastatin for lipid management and plaque stabilization. The patient exhibited clinical improvement (right limb strength recovered to grade 5/5) and was discharged on oral medications with close INR monitoring for warfarin dose adjustment.

## 3. Discussion

This case represents the first reported instance of an NF1 patient harboring the novel NF1 c.2889_2893del (p.Q963Hfs*10) variant concurrently presenting with severe aortic regurgitation, valvular vegetations, left ventricular hypertrophy/enlargement, and left atrial dilation—manifestations not typically part of the natural history of NF1 [8,9]. These severe cardiac structural abnormalities, which ultimately led to cerebral infarction, may be specifically associated with this unique NF1 variant.

The direct cause of cerebral infarction in this case was cardiogenic embolism from dislodged non-infective aortic valve vegetations. However, unlike the typical association, such vegetations are most commonly linked to systemic lupus erythematosus (SLE) and antiphospholipid syndrome (APS), with thromboembolism being a common complication [10,11]. Notably, their occurrence in NF1 patients is exceedingly rare. In this case, common etiologies like SLE, APS, rheumatic heart disease, and Behçet’s disease were excluded. Therefore, the mechanism underlying vegetation formation in the context of NF1 becomes a key question.

The formation of such vegetations may be linked to the unique pathological background of NF1. NF1 gene mutations impair neurofibromin function, which relieves Ras inhibition and causes excessive activation of the RAS/MAPK pathway. This pathway hyperactivation drives abnormal proliferation of vascular smooth muscle and intimal hyperplasia [6], ultimately disrupting vascular endothelial function and vascular homeostasis. The endothelium of valve leaflets, which is inherently fragile as it endures high shear stress and bending stress [12], becomes more prone to injury or detachment if endothelial dysfunction occurs on this basis. Endothelial injury upregulates coagulation factors, triggering platelet adhesion and fibrin deposition [13].

Cardiac disease in NF1 patients is relatively rare and typically manifests as pulmonary artery stenosis (usually valvular) [8]. In contrast, the combination of multiple cardiac structural abnormalities presented in this case is extremely rare. The pathological process demonstrates a clear cascade: aortic valve prolapse complicated by severe regurgitation as the initiating factor, causing left ventricular volume overload, which leads to left ventricular dilation and compensatory hypertrophy. Persistent ventricular remodeling subsequently increases left atrial pressure, resulting in secondary left atrial enlargement and functional mitral regurgitation. Notably, within this cascade, valvular endothelial injury directly provides the pathological basis for vegetation formation.

The patient’s PAI-1 4G/5G heterozygous genotype, with its prothrombotic effect, is another noteworthy factor. PAI-1 is a key inhibitor of the fibrinolytic system. The 4G allele results in higher basal transcription levels of PAI-1 compared to the 5G allele [14]. Consequently, individuals with the 4G/5G heterozygous genotype typically have PAI-1 levels intermediate between those of 4G/4G homozygotes and 5G/5G homozygotes, theoretically conferring a degree of hypofibrinolysis. However, a large body of current evidence has demonstrated that the PAI-1 4G/5G heterozygous genotype is not associated with an increased risk of thrombotic events in clinical settings [14,15,16]. Therefore, this heterozygous genotype may not be considered a contributing factor to the formation of cardiac emboli in this patient. WES identified only a pathogenic variant in the *NF1* gene, with no other genetic abnormalities detected—including those associated with procoagulant function. Sanger sequencing of the *SERPINC1*, *PROC*, and *PROS1* genes further confirmed the absence of pathogenic mutations. Laboratory investigations revealed that PT, APTT and plasma Protein C activity were all within normal reference ranges. Based on all current investigative findings, the formation of cardiac emboli mainly attributed to NF1-mediated cardiac structural and developmental abnormalities. Whether the PAI-1 4G/5G heterozygous genotype plays a role in this process warrants further investigation in future studies.

Compared to rare factors such as cardiogenic embolism and inherited fibrinolytic inhibition that cause cerebral infarction in NF1 patients, plexiform neurofibroma (PNF) represents the most characteristic and severe complication of NF1, with an incidence of 30%~60%. Although PNF is a benign tumor, it can lead to severe consequences including disfiguring lesions, persistent pain, and may even form a fatal compression on the spinal cord [17]. Case reports have documented the locally aggressive behavior of PNF and its significant association with cervical deformities [18]. In this case, the patient presented with a left brachial plexus plexiform neurofibroma, accompanied by morphological distortion of the C4-C6 vertebral bodies and cervical kyphosis. Therefore, the patient’s cervical structural abnormalities are most likely attributed to the invasive effects of the left brachial plexus plexiform neurofibroma, supported by the correlation between plexiform neurofibroma localization and vertebral deformity observed on imaging.

To date, the pathogenesis of NF1-associated cardiac diseases has not been fully elucidated. In homozygous Nf1 mutant mouse models, embryonic loss of neurofibromin leads to severe cardiac structural malformations, including global cardiac hypoplasia, disorganized and hypoplastic myocardium, ventricular septal defects, and abnormal valve leaflet formation [19]. Since homozygous NF1 mutation in humans is likely lethal (as observed in mouse models), no surviving human cases with homozygous NF1 mutation have been reported [8]. Junwang Xu et al. generated a mouse model with cardiomyocyte-specific deletion of neurofibromin (Nf1-cKO), finding that these mice developed significant cardiac hypertrophy, progressive cardiomyopathy, and fibrosis in adulthood, accompanied by hyperactivation of Ras and its downstream signaling pathways, indicating a critical role of NF1 in cardiac hypertrophy and dysfunction [20].

It is important to note that NF1-related vasculopathy (e.g., renal/aortic stenosis, coronary artery aneurysms, or cerebrovascular abnormalities) [21,22,23] can not only directly cause organ ischemia or hemorrhage, but also significantly increase the risk of cardiovascular and cerebrovascular events by involving cardiac structures, increasing cardiac load, or inducing cerebrovascular circulatory disorders. Specifically: arterial stenosis (e.g., renal artery) can increase cardiac afterload, precipitating cardiac dysfunction [21]; coronary artery disease (e.g., aneurysms) can cause myocardial ischemia/infarction [22]; and significant cardiac dysfunction can lead to reduced cardiac output, resulting in hypoperfusion-related brain injury [24,25]. Although the cerebral infarction in this case was primarily caused by rare cardiac valvular pathology rather than as a direct consequence of the classic vascular lesions mentioned above, these mechanisms of vascular–cardiac interaction highlight the importance of systematic cardiovascular assessment—including cardiac structure and function—in NF1 patients.

In patients with NF1, the overall prevalence of congenital heart disease ranges from 2.3% to 12.6% [3,4], reflecting methodological biases. Pinna et al. reported 12.6% (likely overestimated, as the sample included only NF1 patients with cardiac evaluations) and Lin et al. 2.3% (likely underestimated, due to lack of systematic cardiac screening in the cohort). In the systemic evaluation of NF1 patients, the importance of screening for cardiovascular-related diseases should be emphasized, extending beyond skin, nervous system, and tumor manifestations [8,9,26]. Many cardiovascular lesions are often occult and can even be missed by cardiologists [27]. NF1-associated vasculopathy may also escape adequate clinical attention due to the lack of obvious symptoms, with some patients remaining asymptomatic throughout life [2,8]. Therefore, it is strongly recommended that all NF1 patients, regardless of current symptoms, undergo timely cardiovascular assessment using objective imaging modalities such as echocardiography or cardiac magnetic resonance imaging (CMR). For patients without obvious cardiovascular lesions on initial assessment, they should also be clearly informed about the long-term NF1-associated cardiovascular risks, and the necessity of establishing a regular follow-up plan (e.g., periodic imaging review) should be emphasized to enable early identification of potential lesions and timely intervention.

## 4. Conclusions

We report the first case of a patient with neurofibromatosis type 1 (NF1) concurrently presenting with severe aortic regurgitation with valvular vegetations, left ventricular hypertrophy/enlargement, left atrial dilation, mitral regurgitation, and a PAI-1 4G/5G heterozygous genotype, ultimately leading to cerebral and splenic infarction. Based on this experience, we emphasize the need for routine cardiac ultrasound examinations in NF1 patients, with a focus on identifying whether there are developmental abnormalities of the cardiac valves. If further examination is necessary, transesophageal echocardiography can be used to focus on valvular morphology and function, ventricular size and wall thickness, and the presence of vegetations. It is also crucial to comprehensively identify multifactorial thrombotic risks, such as the synergistic effect of cardiac structural abnormalities (e.g., vegetations or chamber enlargement) and inherited thrombophilic factors (e.g., SERPINC1 [e.g., p.Arg150His], PROC [e.g., p.Arg229Trp], PROS1 [e.g., p.Arg536Trp], Factor V Leiden [rs6025], and Prothrombin 20210G>A [rs1799963]), and possibly ‘PAI-1 4G/4G homozygosity’ polymorphisms). Given that NF1 patients are inherently prone to vascular anomalies and associated thrombotic risk, this case underscores the necessity of integrating early, proactive vascular–cardiac assessment and multifactorial thrombotic risk stratification into routine NF1 management. Individualized antithrombotic management strategies, guided by comprehensive risk profiling and long-term follow-up, should be formulated within a multidisciplinary framework to mitigate life-threatening thromboembolic risks.

## Figures and Tables

**Figure 1 diagnostics-16-00163-f001:**
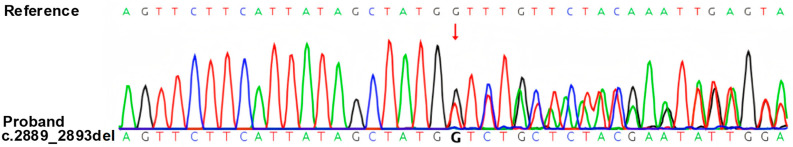
Sequencing comparison of NF1 genes (the upper one is the reference sequence, and the lower one is the proband sequence). The arrow indicates the c.2889_2893del deletion mutation site, which induces a frameshift leading to the amino acid change p.Q963Hfs*10.

**Figure 2 diagnostics-16-00163-f002:**
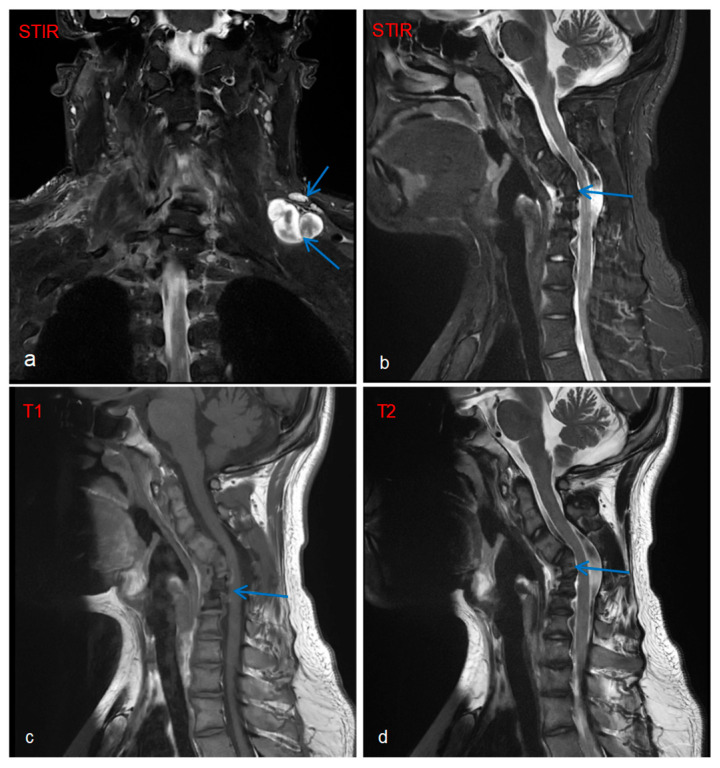
Cervical spine MRI: Target sign observed on fat-suppressed STIR sequence, characterized by peripheral hyperintensity (white) surrounding central hypointensity (black), with the blue arrows indicating the plexiform neurofibroma involving the left brachial plexus (**a**). MRI of the cervical spine: Morphological distortion of C4–C6 vertebral bodies with cervical kyphosis deformity (**b**–**d**).

**Figure 3 diagnostics-16-00163-f003:**
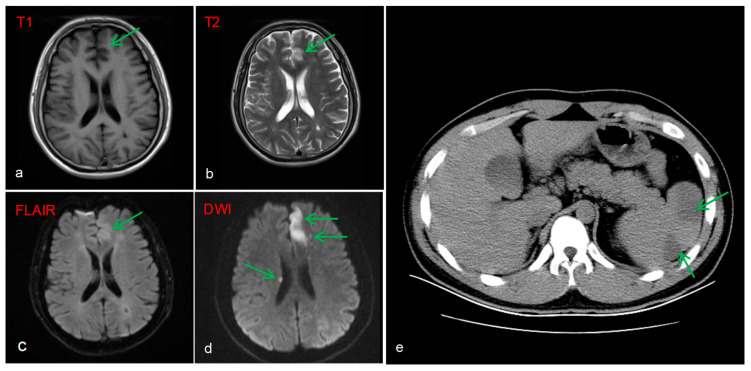
Brain MRI image: Fresh cerebral infarction in the left frontal lobe and right lateral periventricular region (**a**–**d**). Abdominal CT: Radiolucent low-density lesion suggestive of splenic infarction (**e**).

**Figure 4 diagnostics-16-00163-f004:**
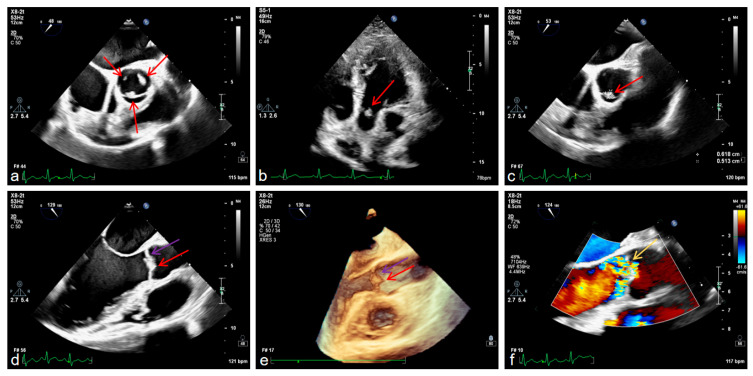
Transesophageal echocardiography (TEE) findings, with red arrows indicating cardiac vegetations and purple arrows denoting aortic valve thickening (**a**–**d**), 3D echocardiography also identifies cardiac vegetations and aortic valve thickening (**e**) and color Doppler echocardiography findings, with yellow arrows indicating blood shunting (**f**).

## Data Availability

The original contributions presented in this study are included in the article. Further inquiries can be directed to the corresponding author.

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
