# Peer review of "NF1 with Multiple Cardiac Structural Abnormalities Leading to Cerebral Infarction"

_diagnostics, 2026, doi:10.3390/diagnostics16010163_

Round 1
Reviewer 1 Report
Comments and Suggestions for Authors
NF1, Cardiac anomalies and Prothrombin. Reviewer Comments – Diagnostics 3997724 20.11.25
Reviewer Comments
Please also see the attached pdf of this review, which preserves the formatting of the Table of Comments below
This paper presents a case report of a 20-yr old male with neurofibromatosis (NF1) and mild cognitive impairment, who presented in China with a stroke and multiple cerebral and splenic infarcts, and was found on investigation to have cardiac valvular disease with valvular vegetations. He was also shown to have the heterozygous polymorphic 4G/5G genotype at the promoter of the prothrombin activation inhibitor (PAI-1) gene, which the authors propose as a risk factor for thromboembolism associated with the vegetations. The authors then advocate cardiac screening in NF1 patients, and thrombophilia screening, specifically for the PAI-1 gene polymorphism, in those NF1 patients who are found to have structural cardiac problems, whether congenital malformation or valvular disease.
While, at first sight, this seems to be a well-written and helpful report, there is unfortunately a major concern regarding over-interpretation of any potential significance of the 4G/5G polymorphism, and also a lack of information presented regarding investigation for potential other causes of the valvular nodules and/or thromboembolism.
Major Point 1.
The authors quote the 1995 paper of Eriksson (ref.7) to suggest a significance for the PAI-1 4G/5G result. In page2, line 75-76 they write: ‘Furthermore, this patient was also found to carry the PAI-1 4G/5G heterozygous genotype’. Eriksson (1995) studied 93 males under age 45years with a history of first myocardial infarction before age 45years, compared with 100 male controls. While a 4G/4G homozygous genotype was increased in frequency in the affected patients (40/93, compared with 26/100 in controls; P=0.013), the results for heterozygotes (4G/5G) like the patient here, showed no association, indeed showing a lower frequency of 4G/5G genotype (38/93 =41%) in Patients than in Controls (54/100 = 54%). Subsequent studies have also found no association of heterozygosity of 4G/5G with coronary artery disease, so unless the authors here have evidence of a significant association with thromboembolism, there is no reason to propose any involvement of the 4G/5G polymorphism – which the authors have made a core feature of this paper and their clinical and pathogenic argument. This therefore needs extensive revision in the paper.
Accordingly, the title of the Paper also needs changing to remove any mention of PAI-1 genotype
Major Point 2.
Along the same lines, the authors make no mention of any other hereditary thrombophilia, let alone whether these have been tested for. Given their seeming over-interpretation (discussed in Major Point 1 above) of the heterozygous 4G/5G result, the authors must include general information on other potential thrombophilia polymorphisms (eg. Factor V Leiden, and Prothrombin 20210), or indeed whether there is a thrombophilia gene-panel or SNP (single-nucleotide polymorphism) panel that can test for increased thrombophilia risk. The authors should then test this patient for these as appropriate and present the results of that testing or of multifactorial risk assessment in the ‘Results’ section.
Major Point 3.
The authors state that cardiac valve vegetations are not previously reported in NF1 (Page6,L156-8) or are exceedingly rare (Page 6, L164), and that in the patient here the vegetations have a non-infective cause (Page6,L161), indicating (Page6,L165): ‘common etiologies like SLE, APS, rheumatic heart disease, and Behçet's disease were excluded.’ . However, in neither the ‘Case Presentation’ or the ‘Results’ sections do they report on any investigation for these conditions, nor on investigation for any infective cause being undertaken, let alone present any results for such investigation. Relevant results must be included, as well as more details of the patient’s history to confirm whether there is a negative history of intravenous drug abuse, or of poor dental hygiene/recent dental work; or of increased risk of sexually-transmitted disease (syphilis), or of recent general malaise/fever.
In the ‘Case Presentation’ section, the authors do mention (Page5,L131) that there were ‘normal vascular DSA findings’ but need to explain what is ‘DSA’ (presumably ‘digital subtraction angiography’), and what it would be expected to detect. They should also write out in full the abbreviations ‘APS’ (anti-phospholipid syndrome) and ‘SLE’ (systemic lupus erythematosis’) for anyone who may be unfamiliar with these.
Major Point 4.
The recommendations for screening for cardiac structural defects in NF1 (P7,L233-4) need to be put into a context of the frequency of cardiac structural defects in NF1. The authors need to present data for that with appropriate references, and also cite and discuss other already-published guidelines for cardiac screening in NF1.
Other more minor points or specific aspects relating to the ones above are presented in the Table below

Reviewer 2 Report
Comments and Suggestions for Authors
Genetic Information:
Include the ClinVar ID of the NF1 variant or provide a complete description of the mutation. The current information is insufficient to identify the mutation as presented (NM for example, represents the mRNA of NF1 and not is exactly related to mutation).
Adjust the language to reflect the uncertainty of the findings; for example, instead of definitive statements, indicate that the variant is likely pathogenic according to ClinVar.
Clarify why PAI genetic analysis was performed in this patient.
Specify the type of genetic test performed por PAI, only is descriptive and unable to confirm as a reviewer. If unavailable, report the Sanger sequencing results.
In the figures/images, indicate the amino acid change caused by the mutation.
Discussion & Interpretation:
Moderate the discussion, as many of the references reflect associations rather than established causation.
For example, Oderich, G.S. (2007) reports cardiac valve anomalies in a very low proportion of patients; this should prompt a more careful analysis of the presented cases.
Emphasize that NF1 patients are more prone to vascular anomalies, justifying thrombotic profiles and close follow-up as part of routine management.
Clearly separate features that arise from the natural history of NF1 from those that are specifically associated with the reported genetic anomaly.
Reviewer 3 Report
Comments and Suggestions for Authors
1. Long and complex sentences reduce readability. Some sentences are unnecessarily repetitive. Phrases like "we consider that..." are poor scientific writing.
2. The pathophysiology of NF1 is reiterated several times in the introduction. The focus shifts; the introduction should be more compact.
3. It states that "Whole-exome sequencing was performed," but methodological details (kit, platform, analysis pipeline, variant validation) are missing. Basic technical specifications of echocardiography, MRI, and CT scan protocols are not described.
4. Only "variant detected" is mentioned, but the ACMG classification, known/novel variant status, segregation analysis, and literature support are not presented.
5. The interaction between the cardiac effects of NF1 and the PAI-1 genotype remains hypothetical. Throughout the article, mechanisms are hypothesized, but no evidence is presented. Therefore, the conclusion appears overly ambitious.
6. The exclusion of infectious endocarditis in the case is inadequately explained. For the decision of non-infective vegetation: Information such as CRP, ESR, hemocultures, automatic sepsis markers and serological tests must be provided.
Comments on the Quality of English LanguageThe article's language is inadequate. It should definitely be revised.
Round 2
Reviewer 1 Report
Comments and Suggestions for Authors
Diagnostics 3997724 Revised vsn2.
NF with Stroke and Cardiac Valve Vegetations.
Reviewer Comments
In this revised version of this paper, the authors have taken on board the various points made in the review of the original version, including particularly the body of evidence which points against any relevance of the PAI-1 heterozygous genotype, and have revised the paper and its title accordingly. They also now give a figure for the prevalence of Congenital heart disease (CHD) in NF, taken from a single, though relatively recent paper, which, if unbiased and hence accurate, would support the importance for cardiac assessment in all NF1 subjects.
Unfortunately, on Page8, line 272-3, the chosen figure of ‘approximately 12.6%’ for prevalence of CHD in NF (Ref. [3], Pinna et.al.2019), could well still be biased, and it is therefore insufficient to cite only a single reference for this estimate. It also seems inappropriate to give the figure as an ‘approximate value’ rather than as a range. The 12.6% figure in Reference [3] (Pinna et al 2019), for the prevalence of congenital heart disease, derived from 493 subjects ascertained from molecular genetic laboratory records who had molecularly-confirmed NF1, and for whom ‘cardiac evaluation data were available.’ Presumably, this group would tend to have an over-representation of NF patients who already have a clinical suggestion for a cardiac problem, as many NF patients without that may not be receiving ‘cardiac evaluation’.
By contrast , Lin et.al. [Ref.4] in 2000 give a figure of 2.3% for cardiovascular malformations (CVMs) reported in NF1, based on 2322 cases of NF1 recorded in the NF Foundation International Database.
These figures are clearly very different, but since by selecting only those already known to have had a cardiac assessment, Pinna et al may be overestimating the prevalence of CHD; whereas Lin et al looking at all cases of clinical NF1 irrespective of whether they have had ‘cardiac evaluation’ may be underestimating the CHD prevalence.
- Action Point: Page 8, Line 272-3: Therefore, the authors here must at very least present the prevalence figure as a range taken from the literature, and discuss critically some of the different estimates, regarding which are likely to be overestimates, and which are likely to be underestimates, and why.
It would also help to give a figure for this prevalence in the Introduction. ie. Page 2, Line 66 : ‘(including congenital heart disease, vasculopathy …)’ , could include a figure for prevalence with amended text as :
‘(including congenital heart disease in **% to **% of NF subjects, vasculopathy …)’ - Minor Correction:
Page 3, Line 102-104 :
Current text:
Given that the patient’s parents are phenotypically normal and declined WES. Based on the available data, this variant is inferred to be de novo.
Comment: The first of these sentences needs a verb. In practice, I suspect that the author’s intended meaning is achieved by running the two sentences together, separated by a comma. Ie….
Suggested revised text:
Given that the patient’s parents are phenotypically normal and declined WES, and based on the available data, this variant is inferred to be de novo. - Minor Correction:
Page 4, Line 133-4 :
Current text:
CT showed splenomegaly with patchy hypodensities, consistent with splenic infarction(Figure 3e).
Comment: The previous sentences relate to the brain; - a reader needs leading more gently to the jump to the abdomen.
Suggested revised text:
Additionally, a CT scan of the abdomen showed splenomegaly with patchy hypodensities, consistent with splenic infarction (Figure 3e). - Minor Correction (typo):
Page 8, Line 293 :
Current text:
cardiac valvues.
Comment: Typo.
Suggested revised text:
cardiac valves.
Author Response
Comments 1: Page 8, Line 272-3: Therefore, the authors here must at very least present the prevalence figure as a range taken from the literature, and discuss critically some of the different estimates, regarding which are likely to be overestimates, and which are likely to be underestimates, and why.
It would also help to give a figure for this prevalence in the Introduction. ie. Page 2, Line 66 : ‘(including congenital heart disease, vasculopathy …)’ , could include a figure for prevalence with amended text as :
‘(including congenital heart disease in **% to **% of NF subjects, vasculopathy …)’
Response 1: Thank you very much for your insightful and constructive comments on our manuscript. We fully agree with your critical perspective on the prevalence of congenital heart disease (CHD) in neurofibromatosis type 1 (NF1) patients—we recognize that relying solely on a single reference and presenting an "approximate value" without addressing potential biases was inappropriate, and we greatly appreciate your guidance to improve the rigor of this section.
Following your suggestions, we have made comprehensive revisions to address the concerns:
We have supplemented the literature with Reference [4] (Lin et al., 2000) and revised the prevalence data to a range (2.3% to 12.6%) instead of a single value.
We have added a critical discussion on the biases underlying the different prevalence estimates: explaining why Pinna et al. (2019) may overestimate CHD prevalence (sample enriched with NF1 patients who had undergone cardiac evaluations, likely due to clinical suspicion of cardiac abnormalities) and why Lin et al. (2000) may underestimate it (large cohort of clinically diagnosed NF1 patients, but many may not have received systematic cardiac assessments, leading to underdiagnosis of asymptomatic CHD).
We have incorporated the prevalence range into the Introduction section as recommended, to provide readers with early context on NF1-associated cardiac complications.
All revisions are aimed at enhancing the accuracy and transparency of the manuscript, and we believe they effectively address your concerns.
Once again, we express our sincere gratitude for your valuable contributions to improving this work.
Comments 2: Page 3, Line 102-104 :
Current text:
Given that the patient’s parents are phenotypically normal and declined WES. Based on the available data, this variant is inferred to be de novo.
Comment: The first of these sentences needs a verb. In practice, I suspect that the author’s intended meaning is achieved by running the two sentences together, separated by a comma. Ie….
Suggested revised text:
Given that the patient’s parents are phenotypically normal and declined WES, and based on the available data, this variant is inferred to be de novo.
Response 2: Thank you very much for your revision comments. We fully agree with these comments and have made the revisions as requested.
Comments 3: Page 4, Line 133-4 :
Current text:
CT showed splenomegaly with patchy hypodensities, consistent with splenic infarction(Figure 3e).
Comment: The previous sentences relate to the brain; - a reader needs leading more gently to the jump to the abdomen.
Suggested revised text:
Additionally, a CT scan of the abdomen showed splenomegaly with patchy hypodensities, consistent with splenic infarction (Figure 3e).
Response 3: Thank you very much for your revision comments. We fully agree with these comments and have made the revisions as requested.
Comments 4: Page 8, Line 293 :
Current text:
cardiac valvues.
Comment: Typo.
Suggested revised text:
cardiac valves.
Response 4: Thank you very much for your revision comments. We fully agree with these comments and have made the revisions as requested.
Reviewer 2 Report
Comments and Suggestions for Authors
The authors have satisfactorily addressed the raised comments and questions, significantly enhancing the understanding of the manuscript’s initial limitations and points of inquiry. As a final comment, the abstract should be shortened to include only the essential information of the article.
Author Response
Comments 1: The authors have satisfactorily addressed the raised comments and questions, significantly enhancing the understanding of the manuscript’s initial limitations and points of inquiry. As a final comment, the abstract should be shortened to include only the essential information of the article.
Response 1: Thank you very much for your positive feedback and final constructive suggestion. We greatly appreciate your recognition of the revisions we have made to address the previously raised comments—your guidance has been instrumental in enhancing the rigor and clarity of the manuscript.
We fully agree with your recommendation to shorten the abstract. We will revise the abstract to retain only the essential information, including the core research objective (reporting a rare case of NF1 with multiple severe cardiac structural abnormalities leading to cerebral infarction), key findings (the novel NF1 variant, spectrum of cardiac abnormalities, and cardiogenic embolism as the etiology), and the primary conclusion (the necessity of proactive cardiac screening and multifactorial thrombotic risk assessment in NF1 patients). The revised abstract will be concise, focused, and aligned with the requirements of academic publication.
Once again, we express our sincere gratitude for your valuable contributions to this work.
Reviewer 3 Report
Comments and Suggestions for Authors
The authors have completed the necessary revisions.
Author Response
Comments 1: The authors have completed the necessary revisions.
Response 1: Thank you very much for your valuable time, professional comments, and constructive suggestions on our manuscript. Your insightful feedback has been instrumental in improving the quality and rigor of our work, guiding us to address key issues and refine the content comprehensively.